# Resveratrol Inhibits Particulate Matter-Induced Inflammatory Responses in Human Keratinocytes

**DOI:** 10.3390/ijms21103446

**Published:** 2020-05-13

**Authors:** Jung-Won Shin, Hyun-Sun Lee, Jung-Im Na, Chang-Hun Huh, Kyung-Chan Park, Hye-Ryung Choi

**Affiliations:** 1Department of Dermatology, Bundang Hospital, Seoul National University, Seongnam 13620, Korea; spellbound00@hanmail.net (J.-W.S.); hyhssky3@naver.com (H.-S.L.); vividna@gmail.com (J.-I.N.); chhuh@snu.ac.kr (C.-H.H.); gcpark@snu.ac.kr (K.-C.P.); 2Department of Dermatology, College of Medicine, Seoul National University, Seoul 03080, Korea; 3Medical Science Research, Bundang Hospital, Seoul National University, Seongnam 13605, Korea

**Keywords:** air pollution, aryl hydrocarbon receptor, oxidative stress, particulate matter, resveratrol, skin inflammation

## Abstract

Particulate matter (PM), a major air pollutant, is a complex mixture of solid and liquid particles of various sizes. PM has been demonstrated to cause intracellular inflammation in human keratinocytes, and is associated with various skin disorders, including atopic dermatitis, eczema, and skin aging. Resveratrol is a natural polyphenol with strong antioxidant properties, and its beneficial effects against skin changes due to PM remain elusive. Therefore, in the present study, we investigated the effect of resveratrol on PM-induced skin inflammation and attempted to deduce the molecular mechanisms underlying resveratrol’s effects. We found that resveratrol inhibited PM-induced aryl hydrocarbon receptor activation and reactive oxygen species formation in keratinocytes. It also suppressed the subsequent cellular inflammatory response by inhibiting mitogen-activated protein kinase activation. Consequentially, resveratrol reduced PM-induced cyclooxygenase-2/prostaglandin E2 and proinflammatory cytokine expression, including that of matrix metalloproteinase (MMP)-1, MMP-9, and interleukin-8, all of which are known to be central mediators of various inflammatory conditions and aging. In conclusion, resveratrol inhibits the PM-induced inflammatory response in human keratinocytes, and we suggest that resveratrol may have potential for preventing air pollution-related skin problems.

## 1. Introduction

Air pollution due to rapid industrialization has become a concerning issue worldwide. Therefore, interest in the adverse effects of particulate matter (PM) on human health is increasing. Particulate matter (PM) is a major harmful air pollutant, and consists of a complex mixture of solid and liquid particles of various sizes [1,2]. PM can be classified as ultrafine (particles with a diameter <0.1 μm:PM_0.1_), fine (particles with a diameter <2.5 μm:PM_2.5_), and coarse (particles with a diameter <10 μm:PM_10_) [3]. PM can penetrate within the respiratory tract beginning with the nasal passage to the alveoli where gas change occurs [3]. Previous studies demonstrated that exposure to PM leads to increased pulmonary inflammation and aggravation of respiratory symptoms [4,5]. PM can exert oxidative stress in the lungs, leading to tissue injury and the infiltration of inflammatory cells [6,7]. Certain surface or soluble components of PM, especially transition metals, can generate reactive oxygen species (ROS) on account of their ability to act as electron donors [8,9]. As well as acting as a source of ROS, PM can also elicit increased ROS generation by exposed cells through an increase of dual oxidase 1 and nicotinamide adenine dinucleotide phosphate oxidase activity [8,10,11]. In addition to respiratory disease, PM has been proven to be related with exacerbation cardiovascular diseases [3]. Previous studies showed that PM can increase plasma viscosity, acute-phase reactants, endothelial dysfunction, and atherosclerotic progression, which are responsible for most cardiovascular diseases [12,13].

Recently, several epidemiological studies have shown that PM also can lead to the development of various skin disorders, including atopic dermatitis, eczema, skin aging, and pigmentation [14,15,16]. The primary mechanisms underlying PM-induced biological effects on the skin, as in other tissues, appear to involve oxidative stress and inflammation [17,18]. PM contains significant amounts of polycyclic aromatic hydrocarbons, which can induce ROS production via aryl hydrocarbon receptors (AhRs) [18,19,20]. Oxidative stress may activate the mitogen-activated protein kinase (MAPK) and nuclear factor kappa b (NF-κB) pathways, leading to the upregulation of cyclooxygenase-2 (COX-2) and prostaglandin E2 (PGE2). In keratinocytes, oxidative stress also induces the expression of various proinflammatory cytokines, including interleukin (IL)–1 alpha, beta, IL-6, IL-8, and MMP-1, which are known to be central mediators of various inflammatory conditions of the skin [21,22].

Resveratrol is a naturally occurring polyphenolic compound, synthesized by plants in response to stressors and fungal infections [23]. Resveratrol neutralizes and inhibits the formation of ROS, and its antioxidant activity is known to be higher than that of vitamin E (65% and 37%, respectively) [24]. It has been proven to have therapeutic and preventive effects against various illnesses, including cancers and cardiovascular diseases [25]. Previous studies have shown that resveratrol also has many beneficial effects on the skin. It provides substantial protection against skin damage from ultraviolet B (UVB) and ultraviolet A (UVA)radiation [26,27,28]. It has been shown to have chemopreventative properties against skin carcinogenesis [29,30]. Furthermore, resveratrol is now being used increasingly in cosmetology owing to its anti-aging properties and potential hypopigmenting activities [31]. It has been demonstrated that topically applied resveratrol penetrates into the skin in a gradient manner, and it is able to maintain its antioxidant and anti-inflammatory efficiency in the skin [31,32]. However, as yet, there have been no reports regarding the effects of resveratrol on PM-induced damage to the skin. Therefore, in this study, we attempted to identify the effects of resveratrol on PM-induced skin inflammation and the underlying molecular mechanisms of action.

## 2. Results

### 2.1. Effect of PM and Resveratrol on Human Keratinocyte Viability

To determine the cytotoxicity of PM and resveratrol in human keratinocytes, the cells were treated with various concentrations of PM (0, 1.25 μg/mL, 3 μg/mL, 6 μg/mL, 12 μg/mL, 25 μg/mL, 50 μg/mL, 100 μg/mL, or 200 μg/mL) and resveratrol (0, 0.01 μg/mL, 0.1 μg/mL, 1 μg/mL, 10 μg/mL, 50 μg/mL, or 100 μM) for 24 h. No significant decreases in cell viability were observed at low PM concentrations up to 25 μg/mL compared to that observed in the untreated control (Figure 1). Therefore, subsequent experiments were performed with 25 μg/mL PM. Resveratrol did not significantly affect cell viability at ≤1 μM over 2 h. These results indicate that resveratrol at the tested concentrations exhibited no cytotoxicity towards keratinocytes.

### 2.2. Resveratrol Reduces Intracellular ROS Levels in PM-Treated Keratinocytes

Given that ROS has been reported to play a critical role in PM-induced inflammation and aging in skin [20,33,34], we measured the generation of intracellular ROS using 2′,7′-dichlorodihydrofluorescein diacetate (DCF-DA) fluorescence dye. As shown in Figure 2, PM increased intracellular ROS levels and resveratrol reduced the levels of PM-induced ROS in a dose-dependent manner in human keratinocytes.

### 2.3. Resveratrol Suppresses PM-Induced COX-2 Expression and PGE2 Generation

We determined the effect of resveratrol on PM-induced COX-2 and PGE2 expression in human keratinocytes. As shown in Figure 3A, the Western blot analysis indicates that pretreatment with resveratrol inhibited PM-induced COX-2 generation in a dose-dependent manner until 0.1μM concentration. In addition, pretreatment with resveratrol dose dependently reduced PM-induced PGE2 expression in the enzyme-linked immunosorbent assay (ELISA) (Figure 3B).

### 2.4. Resveratrol Inhibits Proinflammatory Cytokine Expression

MMP-1 and MMP-9 are known to play a critical role in skin aging via collagen degradation [34,35,36]. They have also been reported to be the proinflammatory cytokines induced by PM in human keratinocytes [17,19,20]. Resveratrol at a concentration of 1 μM significantly reduced the PM-induced increase in MMP-1 and MMP-9 levels (Figure 4A,B). In addition, resveratrol reduced IL-8 levels induced by PM (Figure 4C).

### 2.5. Resveratrol Inhibits PM-Induced MAPK Activation

The Western blot analysis showed that PM activated p38 and c-Jun N-terminal kinase (JNK) (Figure 5). Whereas the activation of JNK was attenuated by resveratrol pretreatment in a dose-dependent manner until the 0.1 μM concentration (Figure 5A), the activation of p38 was not affected by resveratrol.

### 2.6. Resveratrol Inhibits PM-Induced AhR Activation in Keratinocytes

Intranuclear expression of AhR was increased after keratinocytes were exposed to PM, and pretreatment with resveratrol reduced this expression in a dose-dependent manner until the 0.1 μM concentration (Figure 6).

## 3. Discussion

Skin is the outermost organ of the body and various environmental toxic stimuli, such as UV radiation and air pollution, can cause damage to the skin. Evidence indicates that PM induces ROS formation in human keratinocytes and increases COX-2 and proinflammatory cytokines via MAPK activation [20,33,37]. Several studies have also found that this PM-induced inflammatory response may cause skin barrier dysfunction though filaggrin downregulation or tight junction disruption [18,38]. Interestingly, in in vivo animal models, PM has been shown to be able to penetrate into barrier-disrupted skin but not into barrier-intact skin, and induce skin inflammation [17]. These findings indicate that PM may not only trigger an inflammatory skin reaction in dermatologic patients with skin barrier defects (e.g., atopic dermatitis) but also lead to aggravation of barrier dysfunction, thereby creating a vicious cycle.

In the present study, we evaluated the effects of resveratrol on PM-induced skin damage. Our findings suggest that resveratrol suppresses the PM-induced inflammatory response in human keratinocytes. Resveratrol significantly alleviates intracellular ROS levels in PM-treated keratinocytes in a dose-dependent manner. This might be mainly attributed to the radial scavenging action of resveratrol, which has been well documented previously [24]. However, there is the possibility of inhibition of ROS production at higher levels. Interestingly, resveratrol inhibited PM-induced activation of AhR in keratinocytes. AhR is a ligand-activated transcription factor that binds to several exogenous ligands, such as natural plant flavonoids, polyphenolics, as well as synthetic polycyclic aromatic hydrocarbons and dioxin-like compounds [39]. Previous studies have shown that PM-induced inflammatory responses are mediated by the AhR pathway and ROS [17,18,40]. When activated, AhR translocates rapidly from the cytosol to the nucleus. In our study, intranuclear expression of AhR was elevated after keratinocytes were exposed to PM, and pretreatment with resveratrol reduced this expression in a dose-dependent manner. This indicates that resveratrol not only scavenges ROS but also blocks the signaling pathway for ROS generation by PM. Further investigations would be necessary to confirm our findings.

ROS induced by PM activates the MAPK signaling pathways, which, in turn, activates the expression of transcription factors, including c-Jun N-terminal kinase (JNK), dependent activator protein (AP)-1, extracellular signal-regulated kinase (ERK), and p38-dependent NF-κB [18,19]. Additionally, activation of these transcription factors finally leads to COX-2/PGE2 induction and the production of various inflammatory cytokines. In our study, resveratrol ameliorated the PM-induced increase in COX2 and PGE2 production through suppression of the JNK/MAPK pathway. These results indicate that resveratrol can reduce PM-induced skin inflammation and the subsequent barrier dysfunction.

Interestingly, an increase in JNK expression and COX2 production induced by PM was reduced after resveratrol treatment in a dose-dependent manner until the 0.1μM concentration, but this reduction was not convincing at a concentration of 1 μM. Rather, JNK expression and COX2 production showed a further increase at a concentration of 1 μM. This might be due to the biphasic effect of resveratrol. Like other polyphenols, resveratrol can behave either as an antioxidant or prooxidant depending on several parameters, including the dose and microenvironment [41]. Numerous studies showed that resveratrol has biphasic concentration-dependent effects, being an antioxidant at low doses and prooxidant at high doses both in vitro and in vivo [42,43,44,45]. The prooxidant effects of resveratrol are usually followed by phospho-protein kinase B (PKB)/AKR mice thymoma (Akt) downregulation, cellular damage, and apoptosis [41]. Flavin oxidase-generated ROS and cytochrome P450 enzyme CYP2C9-produced ROS appear to be involved in high dosage-associated resveratrol-elicited oxidative damage [46]. Generally, resveratrol, at concentrations between 0.1 and 1µM, is not cytotoxic and exhibits beneficial effect to cells [47,48,49]. Although we also confirmed that resveratrol did not significantly affect keratinocyte viability at ≤1 μM, this result might change in a specific situation like a PM-induced inflammatory condition. The optimal cytoprotective dose of resveratrol needs to be determined through further investigations.

Resveratrol also suppressed MMP-1, MMP-9, and IL-8 production induced by PM. MMP-1 and MMP-9 are well-known proteolytic enzymes that degrade the extracellular matrix of the skin, leading to wrinkle formation and photoaging [35,50]. IL-8 has also been found to be associated with photoaging [51]. These data suggest that resveratrol may be able to prevent PM-induced skin aging. Kim et al. showed that PM can lead to dermal aging via the ROS-induced inflammatory process in fibroblasts co-cultured with keratinocytes [20]. Further investigations on the effect of resveratrol on PM-induced changes in fibroblasts are required.

This study has several limitations. First, we used keratinocytes derived from foreskins, which are not exposed to PM. Although there have been no data on whether there is a difference in response to air pollution depending on the anatomical site from which keratinocytes were derived, it is expected that using keratinocytes in the region that is prone to PM exposure will get more accurate results. However, because getting enough skin samples from the exposed area is practically difficult, excised skins during circumcision were used to obtain keratinocytes in this study. Second, we used ERM CZ120 from Sigma-Aldrich as PM10-like material. ERM CZ120 was prepared by jet-milling of dust collected from a Warsaw road tunnel [52]. Since the composition of PM may differ from region to region, it might be difficult to generalize the results of the present study to various situations in different geographic locations. The most ideal strategy would be analyzing all of the components of the PM and identifying which one constitutes its biological effects [17].

In conclusion, in the present study, we found that resveratrol inhibits the PM-induced inflammatory response in human keratinocytes by inhibiting the AhR/ROS/MAPK pathway.

## 4. Materials and Methods

### 4.1. Materials

Fine dust (PM_10_-like, ERMCZ120), resveratrol (R5010), 4′,6-diamidine-2′-phenylindole dihydrochloride (DAPI, 10236276001), and 2′,7′-dichlorofluorescin diacetate (DCFDA, D6883) were purchased from Sigma (St. Louis, MO, USA). Monoclonal antibodies for the Ah Receptor (AhR, sc133088, 1:100) and COX-2 (sc19999, 1:50) were purchased from Santa Cruz Biotechnology (Dallas, TX, USA). Monoclonal antibodies for COX-1 (#160110, 1:200) were purchased from Cayman Chemical (Ann Arbor, MI, USA). Polyclonal antibodies for phospho-SAPK/JNK (Thr183/Tyr185, #9251, 1:500), SAPK/JNK (#9252, 1:1000), phospho-p38 MAPK (Thr180/Tyr182, #9211, 1:500), p38 MAPK (#9212, 1:1000), beta-actin (#4967, 1:1000), horseradish peroxidase (HRP)-linked antibodies recognizing mouse IgG (#7076, 1:1000), and rabbit IgG (#7074, 1:1000) were purchased from Cell Signaling Technology (Danvers, MA, USA).

### 4.2. Cell Culture

Normal human epidermal keratinocytes were separated from foreskin samples obtained from three donors of the same age (11 years old) following circumcision procedures. The cells were cultured according to official guidelines of the Institutional Review Board (IRB Approval No. B-1603/340-309) (07 April 2016) of Seoul National University’s Bundang Hospital. We treated skin specimens based on a previous report [53] and modified them using thermolysin. Keratinocytes were cultured in Keratinocyte Growth Medium (#00192060, Lonza, Wakersville, MD, USA) in 5% CO_2_ at 37 °C. Early-passage keratinocytes (passage 3) were used for experiments.

### 4.3. Cell Viability Assay

To evaluate the effects of PM and resveratrol on cell viability, keratinocytes (8000 cells per well) were seeded into 96-well plates. After serum starvation for 24 h, the cells were treated with PM_10_ or resveratrol. The cells were then incubated for another 24 h, and cell viability was determined using a Cell Counting Kit-8 (CCK-8, CK04, Dojindo Molecular Technologies, Inc., Rockville, MD, USA). Absorbance was measured at 450 nm using an Epoch2 microplate reader (BioTek, Winooski, VT, USA). Cell viability was calculated using absorbance values and data were normalized to the untreated control.

### 4.4. Cellular Reactive Oxygen Species (ROS) Assay

Keratinocytes (120,000 cells per chamber) were first seeded onto chamber slides (#154852, Thermo Fisher Scientific, Waltham, MA, USA). After incubation overnight and following 2 h of serum starvation, resveratrol was added, and then 6 h later, PM_10_ was added. After another overnight incubation, the cells were incubated with DCF-DA (10 µM) at 37 °C for 30 min. The slides were washed with keratinocyte basal media three times and mounted. Digital images were acquired using a confocal laser scanning microscope (LSM710, Zeiss, Oberkochen, Germany) and analyzed using ImageJ (NIH, Bethesda, MD, USA).

### 4.5. Immunohistochemistry

Keratinocytes were cultured and treated in the same manner as that described for the cellular ROS assay. The cells were fixed with cold methanol for 15 min and washed with phosphate-buffered saline (PBS) three times, followed by blocking with 5% normal goat serum (ab7481, abcam, Cambridge, MA, USA). The cells were again washed in PBS for 90 min, and then incubated with primary antibody diluted in 5% normal goat serum at 4 °C overnight. After washing cells with PBS three times, the cells were incubated with goat anti-mouse IgG-Alexa 488 (1:500 dilution, A11001, Thermo Fisher Scientific, Waltham, MA, USA) in 5% normal goat serum for 90 min at room temperature. The culture slides were rinsed with PBS three times, stained with DAPI (1 ug/mL), and mounted. Digital images were acquired using a confocal laser scanning microscope (Zeiss, Oberkochen, Germany) and analyzed using ImageJ (NIH).

### 4.6. Inflammatory Cytokine and Secretory Protein Analysis

Keratinocytes (150,000 cells per well) were seeded onto 6-well plates. After incubation overnight, and 2 h of serum starvation, resveratrol was added, and 6 h later, PM_10_ was added. The cells were then incubated overnight again, and cell culture supernatants for further analyses were collected. The quantification of PGE2 was performed by using an ELISA Kit (#514010, Cayman Chemical, Ann Arbor, MI, USA). In addition, a customized Magnetic Luminex Kit (LXSAHM, R&D Systems, Minneapolis, MN, USA) was used for the quantification of MMP1, MMP9, and IL8. Luminex is an array-based multiplex ELISA system for simultaneous quantitative measurement of multiple cytokines, growth factors, proteases, soluble receptors, and other proteins in a wide variety of sample types.

### 4.7. Western Blot Analysis

Protein samples were prepared using an AllPrep DNA/RNA/Protein Mini Kit (#80004, QIAGEN Inc., Germantown, MD, USA). Samples and protein ladders (#26619, Thermo Fisher Scientific) were separated on Invitrogen Bolt 10% Bis-Tris Plus gels (NW00100BOX, Thermo Fisher Scientific) and then transferred to Immobilon-P PVDF membranes (IPVH00010, Millipore, Billerica, MA, USA), which were blocked with 5% nonfat dry milk (#1706404, Bio-Rad Laboratories, Hercules, CA, USA) in Tris buffered saline (TBS) solution containing 0.1% Tween 20. The blots were then incubated with the primary antibodies at 4 °C overnight. Membrane-bound primary antibodies were detected using HRP-linked secondary antibodies (Cell Signaling Technology, Danvers, MA, USA) and a chemiluminescent substrate (#1705060, Bio-Rad Laboratories, Hercules, CA, USA). Digital images of the blotted membranes were obtained using ChemiDoc XRS+ Imaging Systems (Bio-Rad Laboratories, Inc. city, state abbreviation if USA or Canada, country) and analyzed using ImageJ 1.52p bundled with 64-bit Java 1.8.0_172 (NIH, Bethesda, MD, USA).

### 4.8. Statistical Analysis

All data are presented as the mean ± standard deviation. The mean values were calculated based on data from three independent experiment that used the keratinocytes from different donors and were conducted on separate days. The significance of differences between treatments was assessed using the Student’s *t*-test in Excel.

## 5. Conclusions

We showed that resveratrol inhibits PM-induced ROS formation in keratinocytes and also suppresses the subsequent cellular inflammatory response via inhibition of MAPK activation. Consequentially, resveratrol reduced the PM-induced COX2/PGE2 expression and proinflammatory cytokine expression associated with skin barrier dysfunction and aging. Therefore, we suggest that resveratrol may have a potential role in the prevention of air pollution-related skin problems.

## Figures and Tables

**Figure 1 ijms-21-03446-f001:**
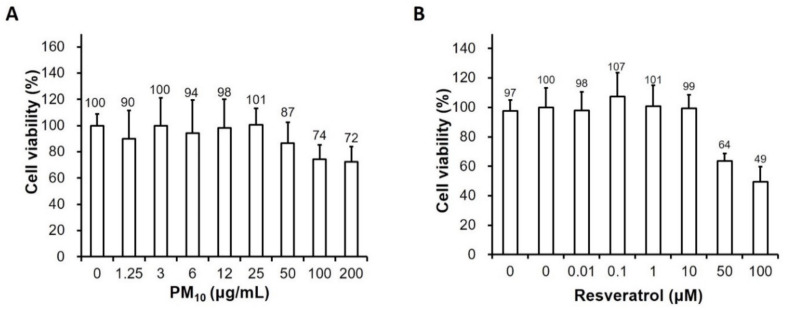
Effects of particulate matter and resveratrol on human keratinocyte viability. Viability of keratinocytes treated with (**A**) PM_10_ (0, 1.25 μg/mL, 3 μg/mL, 6 μg/mL, 12 μg/mL, 25 μg/mL, 50 μg/mL, 100 μg/mL, or 200 μg/cm^2^) or (**B**) resveratrol (0, 0.01 μg/mL, 0.1 μg/mL, 1 μg/mL, 10 μg/mL, 50 μg/mL, or 100 μM) for 24h measured using the water-soluble tetrazolium salt assay. Data are presented as the mean ± standard deviation of three independent experiments (*n* = 3). The ratio of surviving cells is relative to the untreated control. PM, particulate matter.

**Figure 2 ijms-21-03446-f002:**
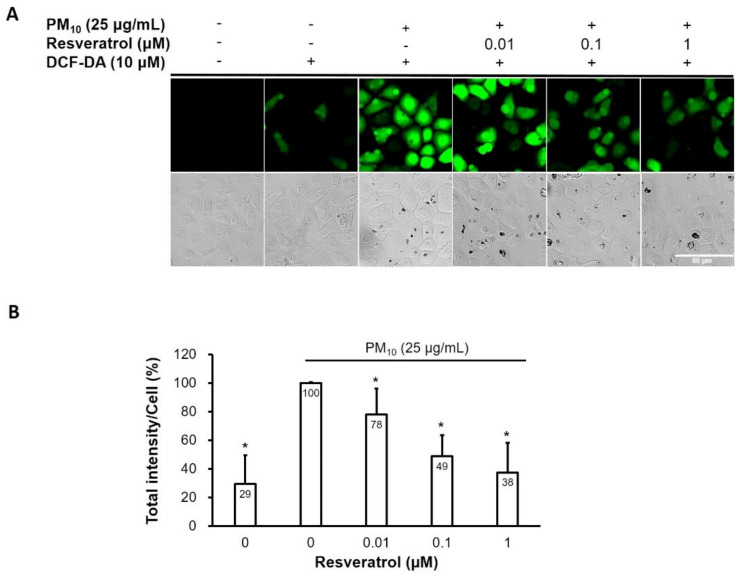
ROS-reducing effect of resveratrol on human keratinocytes following PM treatment. (**A**) Confocal microscopic images of the DCF-DA assay (ROS; green) (**B**) Relative intracellular ROS fluorescence intensity. Data are presented as the mean ± standard deviation of three independent experiments (*n* = 3). * *p* < 0.05 vs. PM-treated, resveratrol-untreated samples. ROS, reactive oxygen species, PM, particulate matter, DCF-DA, 2′,7′-dichlorodihydrofluorescein diacetate.

**Figure 3 ijms-21-03446-f003:**
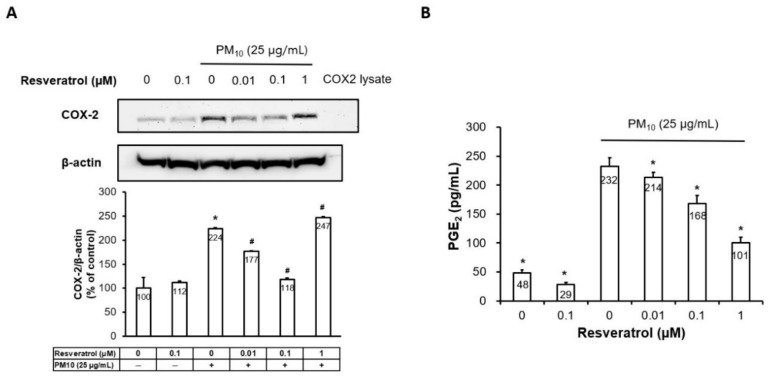
Effect of resveratrol on PM-induced COX-2 expression and PGE2 generation in human keratinocytes. (**A**) The protein expression of COX-2 determined by Western blot and densitometry. Equal amounts of protein loading were verified using an anti-actin antibody, (**B**) PGE2 generation measured by ELISA, Data are presented as the mean ± standard deviation of three independent experiments (*n* = 3). * *p* < 0.05 and ^#^
*p* < 0.01 compared with untreated cells and PM-treated cells, respectively. PM, particulate matter, COX-2, cyclooxygenase-2, PGE2, prostaglandin E2, ELISA, enzyme-linked immunosorbent assay.

**Figure 4 ijms-21-03446-f004:**
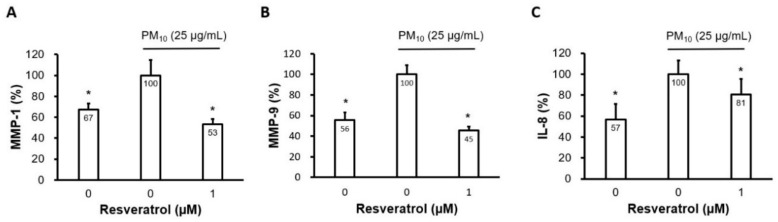
Effect of resveratrol on the production of proinflammatory cytokines in human keratinocytes following PM treatment. (**A**) Protein levels of MMP-1, (**B**) MMP-9, and (**C**) IL-8 were determined using ELISA. Cells were pretreated with resveratrol (1 μM) for 6 h and subsequently incubated with PM_10_ (25 μg/cm^2^) for 24 h. Keratinocyte culture supernatants were collected and used for further analyses. Data are presented as the mean ± standard deviation of three independent experiments (*n* = 3). * *p* < 0.05 vs. PM-treated, resveratrol-untreated control, PM, particulate matter, IL, interleukin, MMP, matrix metalloproteinase, ELISA, enzyme-linked immunosorbent assay.

**Figure 5 ijms-21-03446-f005:**
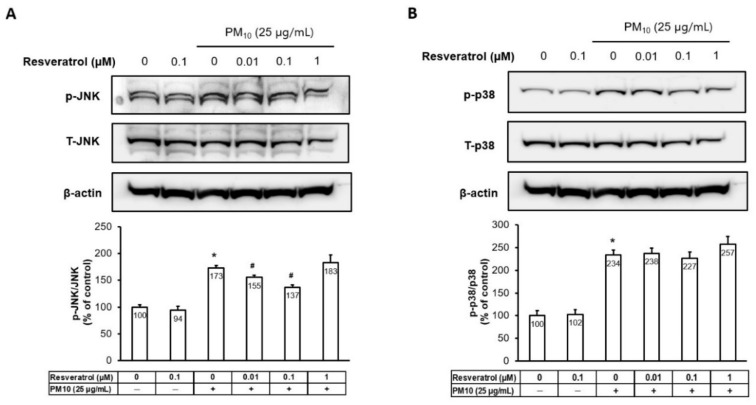
Effects of resveratrol on transcription factor activator protein-1 components and p38 MAPK phosphorylation in human keratinocytes following PM treatment. (**A**) Protein levels of JNK and (**B**) p38 MAPK were analyzed by Western blotting and densitometry. Equal amounts of protein loading were confirmed using JNK, p38, and actin antibodies. * *p* < 0.05 and ^#^
*p* < 0.05 compared with untreated cells and PM-treated cells, respectively. PM, particulate matter; JNK, Jun N-terminal kinase; MAPK, mitogen-activated protein kinase; p, phosphorylated; T, total.

**Figure 6 ijms-21-03446-f006:**
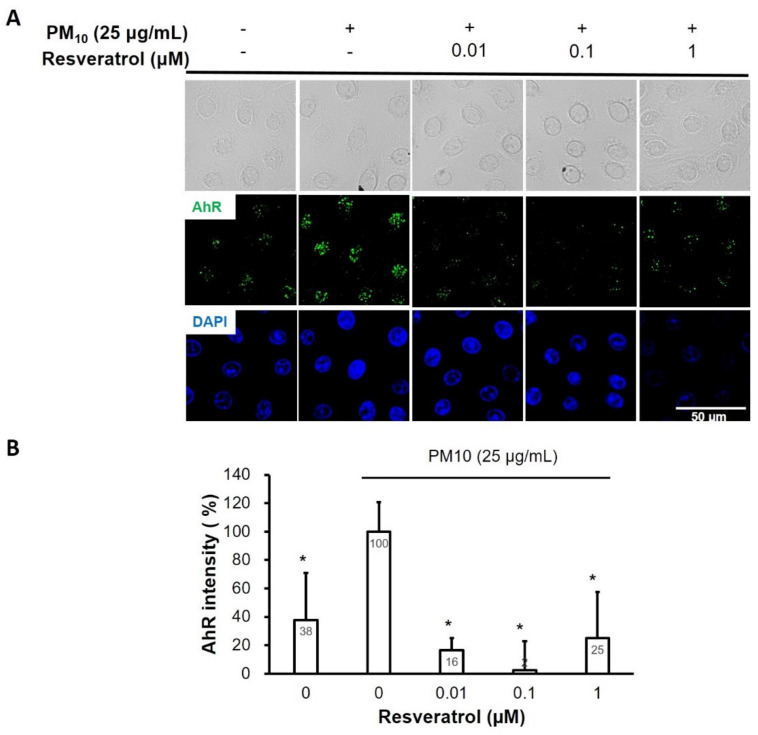
Effect of resveratrol on AhR expression in human keratinocytes following PM treatment. (**A**) Confocal microscopic images of AhR expressions in keratinocytes (AhR; green, nucleus; blue). (**B**) Relative AhR intensity. Data are presented as the mean ± standard deviation of three independent experiments (*n* = 3). * *p* < 0.01 vs. PM-treated, resveratrol-untreated samples. AhR, aryl hydrocarbon receptor; DAPI, 4,6-diamidino-2-phenylindole; PM, particulate matter.

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
