# Peer review of "Resveratrol Inhibits Particulate Matter-Induced Inflammatory Responses in Human Keratinocytes"

_ijms, 2020, doi:10.3390/ijms21103446_

Round 1
Reviewer 1 Report
This is an interesting study, detailing at biochemical and molecular level the protective effects offered by resveratrol against airborne particulat matter-induced oxidative stress and inflammation. The experiments are accurately planned and executed and the results are presented in a clear way. The results are of interest in the specific field insofar as they add to previous existing data concerning lung oxidative injury and inflammation induced by PM.
Author Response
This is an interesting study, detailing at biochemical and molecular level the protective effects offered by resveratrol against airborne particulate matter-induced oxidative stress and inflammation. The experiments are accurately planned and executed and the results are presented in a clear way. The results are of interest in the specific field insofar as they add to previous existing data concerning lung oxidative injury and inflammation induced by PM.
--> Thank you for your valuable comments on our manuscript. According to your suggestion, now we added contents of PM-induced oxidative injury and inflammation of lung in the Introduction (Line 37-47) as follows.
: PM can penetrate within the respiratory tract beginning with the nasal passage to the alveoli where gas change occurs. Previous studies demonstrated that exposure to PM leads to increased pulmonary inflammation and aggravation of respiratory symptoms. PM can exert oxidative stress in lung leading to tissue injury and infiltration of inflammatory cells. Certain surface or soluble components of PM, especially transition metals, can generate ROS on account of their ability to act as electron donors. As well as acting as a source of ROS, PM can also elicit increased ROS generation by exposed cells through increase of dual oxidase 1 and nicotinamide adenine dinucleotide phosphate oxidase activity. In addition to respiratory disease, PM has been proven to be related with exacerbation cardiovascular diseases. Previous studies showed that PM can increase plasma viscosity, acute-phase reactants, endothelial dysfunction, and atherosclerotic progression which are responsible for most cardiovascular diseases.
Reviewer 2 Report
This is a study using primary human keratinocyte cultures to evaluate the anti-inflammatory effects of resveratrol after stimulation with particulate matter (to mimic skin inflammation due to air pollution). Specific comments are as follows:
Keratinocyte cultures: How many different donors were used? At what passages were cells included in experiments? Are there any important known differences in the keratinocytes derived from foreskin and those in skin at sites more likely to be exposed to air pollution?
All experiments: How many times was each assay repeated, i.e., how representative are the results that are shown? If an n=3 was indicated, does this reflect triplicate assay values from the same samples, or individual values from triplicate culture wells, or what exactly?
Western blots: The method mentions analyses using ImageJ. However, both figures which show Western blots (Figure 3A and Figure 5) do not indicate that any semi-quantitative densitometry was performed. This would be helpful in the presentation of results.
Page 4, lines 101-108: This paragraph is redundant with Methods section 4.4.
Figure 3A: A reduction of COX-2 expression is not convincing at 1 uM resveratrol.
Page 5, lines 126-127: This sentence needs a little grammatical help.
Figure 4: The text refers to Figure 4A, B, and C, but there are no such labels in the figure. The results text and/or figure legend should explicitly state that the samples tested were supernatants from the keratinocyte cultures. The figure legend states that an ELISA was used, but the methods stat that a Luminex assay was used. Why is there no mention in the results of the other proteins assayed via Luminex (as stated in the methods): IL-1 alpha, IL-1 beta, IL6, MMP-2 and MMP-13? If there were no differences, those would be important data to discuss.
Figure 5: The results state that resveratrol pretreatment reduced p38 and JNK in a dose-dependent manner, however, the Western blot is really not very convincing. Again, densitometry would be helpful. Also, is it common to see two bands when detecting JNK in Westerns? The first line of the figure legend is wrong and likely was a copy-paste error in the manuscript.
Discussion: A good discussion, but there is no deliberation of the limitations of this study.
Author Response
This is a study using primary human keratinocyte cultures to evaluate the anti-inflammatory effects of resveratrol after stimulation with particulate matter (to mimic skin inflammation due to air pollution). Specific comments are as follows:
- Thank you for your careful, constructive, and valuable comments to help improve the quality of our manuscript. Now, our manuscript has been revised according to your suggestions.
Keratinocyte cultures: How many different donors were used? At what passages were cells included in experiments? Are there any important known differences in the keratinocytes derived from foreskin and those in skin at sites more likely to be exposed to air pollution?
- In this study, cell culture was performed using keratinocytes from three different donors of the same age (11 years old – in Korea, circumcision is most frequently performed in early teenagers). In addition, early passage cells (passage 3) were used for all the experiments. These contents were added in Experimental section 4.2 (Line 250-251).
- Thank you for pointing the important issue regarding possible difference of keratinocytes from different anatomic sites. Previous studies showed that keratinocytes from different anatomical sites have site-specific characteristics such as immune response against pathogen or differentiation pattern (Israr, et al. Mol Med 2018;24;23, Compton, et al. 1998;64:45-53). Therefore, it can be expected that using keratinocytes in the region that is prone to PM exposure will get more accurate results in the present study. However, as you know, getting enough skin samples from exposed area is difficult. So far, there have been no data regarding differences between the keratinocytes derived from non-exposed area and exposed area. Further investigation is needed to clarify this issue. These contents were now added in the Discussion (Line 215 -220) as below.
: First, we used keratinocytes derived from foreskins which are not exposed to PM. Although there have been no data on whether there is a difference in response to air pollution depending on the anatomical site form which keratinocytes were derived, it is expected that using keratinocytes in the region that is prone to PM exposure will get more accurate results. However, because getting enough skin samples from exposed area is practically difficult, excised skins during circumcision were used to obtain keratinocytes in this study.
All experiments: How many times was each assay repeated, i.e., how representative are the results that are shown? If an n=3 was indicated, does this reflect triplicate assay values from the same samples, or individual values from triplicate culture wells, or what exactly?
- Three independent experiments using the keratinocytes from each donor were performed. This content was now added in the Experimental session 4.8 (Line 305-307)
Western blots: The method mentions analyses using ImageJ. However, both figures which show Western blots (Figure 3A and Figure 5) do not indicate that any semi-quantitative densitometry was performed. This would be helpful in the presentation of results.
- Thank you for your valuable comments. The data of densitometry were added in the Results (Figure 3, Figure 5).
Page 4, lines 101-108: This paragraph is redundant with Methods section 4.4.
- We agree with you. As you suggested, the redundant paragraph was deleted (Line 96).
Figure 3A: A reduction of COX-2 expression is not convincing at 1 uM resveratrol.
- Thank you for pointing this out. Actually, it is an important point and needs to be discussed, but was missing in the previous version. As you mentioned, some of our data including COX-2 and p-JNK expression showed that resveratrol can reduce proinflammatory process in a dose dependent manner until 0.1µM, but rather increase the expression of these molecules in the concentration of 1µM. This might be due to the biphasic effect of Resveratrol. As you know, like other polyphenols, resveratrol exhibits biphasic dose-dependent effects. At low concentrations, resveratrol acts as an antioxidant which can protect from DNA damage and oxidative stress. On the other hand, at high concentrations, resveratrol acts as a pro-oxidant promoting DNA damage while increasing oxidative stress. Generally, resveratrol, at concentrations between 0.1 and 1µM is not cytotoxic and exhibits beneficial effect to cells. Although we also confirmed that resveratrol did not significantly affect keratinocyte viability at ≤ 1 μM, this result might change in a specific situation like PM-induced inflammatory condition. We believe our results provide a basic clue to determining the effective and safe dose of resveratrol in reducing the PM-induced inflammatory response in keratinocytes. And through further investigations, the optimal cytoprotective dose of resveratrol might be determined. These contents were now added in the Discussion (Line 196-207).
Page 5, lines 126-127: This sentence needs a little grammatical help.
- The grammatical errors were checked and corrected as below (Line 122-124)
:MMP-1 and MMP-9 are known to play a critical role in skin aging via collagen degradation. They have also been reported to be the proinflammatory cytokines induced by PM in human keratinocytes
Figure 4: The text refers to Figure 4A, B, and C, but there are no such labels in the figure. The results text and/or figure legend should explicitly state that the samples tested were supernatants from the keratinocyte cultures. The figure legend states that an ELISA was used, but the methods stat that a Luminex assay was used. Why is there no mention in the results of the other proteins assayed via Luminex (as stated in the methods): IL-1 alpha, IL-1 beta, IL6, MMP-2 and MMP-13? If there were no differences, those would be important data to discuss.
- Thank you for your careful review. Now the subtitle of Figure 4 - (A) Protein levels of MMP-1, (B) MMP-9, and (C) IL-8 levels were added in the figure legend (Line 130). In addition, a comment about supernatants was inserted in the figure legend of Figure 4 (Line 132)
- The Luminex is trademark of an array-based multiplex ELISA system for simultaneous quantitative measurement of multiple cytokines, growth factors, proteases, soluble receptors, and other proteins in a wide variety of sample types. This content was added in the Experimental section (Line 286-290).
- Actually, we only tested protein levels of MMP-1, MMP-9, and IL8 that were found to be affected resveratrol in our previous study using cells other than keratinocytes (unpublished data). The other proteins (IL-1, IL6, MMP-2, MMP-13) were used in the previous study and described as a mistake. We apologize for the confusion. Now, the contents were deleted in the manuscript (Line 288).
Figure 5: The results state that resveratrol pretreatment reduced p38 and JNK in a dose-dependent manner, however, the Western blot is really not very convincing. Again, densitometry would be helpful. Also, is it common to see two bands when detecting JNK in Westerns? The first line of the figure legend is wrong and likely was a copy-paste error in the manuscript.
- That’s a very important point. As you suggested, the results of densitometry were now included in the manuscript (Figure 3 and Figure 5). The results of COX2 and p-JNK were same as we described (reduced by resveratrol dose-dependently except at the concentration of 1µM – biphasic effect). However, quantitative data on p38 showed different results – not affected by resveratrol treatment. So, we concluded that resveratrol exhibits anti-inflammatory effects mainly through inhibition of the JNK pathway among the cellular stress response of keratinocytes induced by PM. And related contents in the Results and Discussion were corrected (Line 139-141, Line 192-193).
- As far as we know, p-JNK generally include JNK1/JNK2 (Thr183, Tyr185) and have two bands in Western blot analysis.
- The figure legend was corrected (afzelin à resveratrol, Line 144).
Discussion: A good discussion, but there is no deliberation of the limitations of this study.
- Thank you. Now, the limitations of this study were added in the Discussion as below (Line 215-226)
: This study has several limitations. First, we used keratinocytes derived from foreskins which are not exposed to PM. Although there have been no data on whether there is a difference in response to air pollution depending on the anatomical site form which keratinocytes were derived, it is expected that using keratinocytes in the region that is prone to PM exposure will get more accurate results. However, because getting enough skin samples from exposed area is practically difficult, excised skins during circumcision were used to obtain keratinocytes in this study. Second, purchased ERM CZ120 from Sigma-Aldrich and used as PM10-like material. The approach adopted for the preparation of ERM CZ120 was based on jet-milling of dust collected from a Warsaw road tunnel. Since the composition of PM may differ from region to region, it might be difficult to generalize the results of the present study to various situations in each geographic location. The most ideal strategy would be analyzing all of the components of the PM and identifying which one constitutes its biological effects.
Reviewer 3 Report
This manuscript addresses an actual thematic on the application of natural products with health benefits, for air-pollution related diseases, which are a major concern in highly populated and industrialised countries.
Nevertheless, the manuscript in its present form, has several shortcomings which need to be highly improved to ensure significance.
Detailed comments
-The statistical approach of this set of experimental data must be detailed. More importantly, from the description and data presentation included in the manuscript, the results are mainly representative of one independent experiment with three replicates. This means that the indication of N=3 used by the authors, might not be correct and this should be clarified. If this is the case, the results are not significant and should represent only indications that need to be validated with more experiments N≥3.
-In the material and methods the description on how the primary cultures were prepared and characterizes is lacking (in 4.2). Also, to ensure a representative N the authors must state how many skin donors the authors use to prepare the primary cell cultures. Moreover, in this MM section the authors must state which cell passage was used for each independent experiment.
-In figure 3 A and 5 the western results presented, in the current form are simply qualitative. A representative quantitative description including statistical significance data, should be performed and included.
Author Response
This manuscript addresses an actual thematic on the application of natural products with health benefits, for air-pollution related diseases, which are a major concern in highly populated and industrialised countries.
Nevertheless, the manuscript in its present form, has several shortcomings which need to be highly improved to ensure significance.
- Thank you for your careful, constructive, and valuable comments to help improve the quality of our manuscript. Now, our manuscript has been revised according to your suggestions.
Detailed comments
-The statistical approach of this set of experimental data must be detailed. More importantly, from the description and data presentation included in the manuscript, the results are mainly representative of one independent experiment with three replicates. This means that the indication of N=3 used by the authors, might not be correct and this should be clarified. If this is the case, the results are not significant and should represent only indications that need to be validated with more experiments N≥3.
- Thank you for pointing this out. In this study, 3 independent experiments using the keratinocytes from each donor were performed. This content was now added in the Experimental session 4.8 (Line 305-307)
-In the material and methods the description on how the primary cultures were prepared and characterizes is lacking (in 4.2). Also, to ensure a representative N the authors must state how many skin donors the authors use to prepare the primary cell cultures. Moreover, in this MM section the authors must state which cell passage was used for each independent experiment.
* The process of keratinocyte culture was now described more in detail in Experimental section 4.2 (Line 246, 250-251). Cell culture was performed using keratinocytes from three different donors of the same age (11 years old – in Korea, circumcision is most frequently performed in early teenagers). In addition, early passage cells (passage 3) were used for all the experiments.
-In figure 3 A and 5 the western results presented, in the current form are simply qualitative. A representative quantitative description including statistical significance data, should be performed and included.
- That’s a very important point. As you suggested, the results of densitometry were now included in the manuscript (Figure 3 and Figure 5). The results of COX2 and p-JNK were same as we described (reduced by resveratrol dose-dependently except at the concentration of 1µM – biphasic effect). However, quantitative data on p38 showed different results – not affected by resveratrol treatment. So, we concluded that resveratrol exhibits anti-inflammatory effects mainly through inhibition of the JNK pathway among the cellular stress response of keratinocytes induced by PM. And related contents in the Results and Discussion were corrected (Line 139-141, Line 192-193).